# FRAMEWORK OF THOUGHTS: A FOUNDATION FRAMEWORK FOR DYNAMIC AND OPTIMIZED REASONING BASED ON CHAINS, TREES, AND GRAPHS

## ABSTRACT

Prompting schemes such as Chain of Thought, Tree of Thoughts, and Graph of Thoughts can significantly enhance the reasoning capabilities of large language models. However, most existing schemes require users to define static, problem-specific reasoning structures that lack adaptability to dynamic or unseen problem types. Additionally, these schemes are often under-optimized in terms of hyper-parameters, prompts, runtime, and prompting cost. To address these limitations, we introduce Framework of Thoughts (FoT)–a general-purpose foundation framework for building and optimizing dynamic reasoning schemes. FoT comes with built-in features for hyperparameter tuning, prompt optimization, parallel execution, and intelligent caching, unlocking the latent performance potential of reasoning schemes. We demonstrate FoT's capabilities by implementing three popular schemes–Tree of Thoughts, Graph of Thoughts, and ProbTree–within FoT. We empirically show that FoT enables significantly faster execution, reduces costs, and achieves better task scores through optimization. We release our codebase to facilitate the development of future dynamic and efficient reasoning schemes.

## 1 INTRODUCTION

*Large language models* (LLMs) have become popular for various problem-solving and reasoning tasks such as mathematical or logical reasoning (Cobbe et al., 2021), task planning (Shridhar et al., 2021), or multi-hop question-answering (Yang et al., 2018; Trivedi et al., 2022). It has been shown that, similar to humans, LLMs' accuracy on these tasks improves significantly when LLMs generate a step-by-step thought process before concluding a final answer (Wei et al., 2022; Kojima et al., 2022). Multiple prompting schemes have been proposed to elicit such thought processes in LLMs: *Chain of Thought* (CoT) (Wei et al., 2022) and *zero-shot* CoT (Kojima et al., 2022) include examples of desired thought sequences or an instruction to think step by step in the prompt. Later schemes such as *Tree of Thoughts* (ToT) (Yao et al., 2023) or *Graph of Thoughts* (GoT) (Besta et al., 2024a) involve multiple prompts organizing the LLM's thoughts into more complex tree or graph structures rather than linear chains. Several other prompting schemes based on chains, trees, and graphs have been proposed. However, most of these prompting schemes come with some key limitations.

**Limitation #1: The prompting schemes rely on manually-defined and static graph structures.** The vast majority of the schemes are not fully automatic (refer to Besta et al., 2024b), requiring users to manually specify task-specific prompts and graph structures that define how to decompose and solve a problem type. The graph structure then remains static during execution and the LLM only fills in some specific thoughts related to the problem. This typically prevents generalizability to previously unseen or dynamic problems where the ideal reasoning structure is not known a-priori but must be actively discovered for every problem instance (Zhou et al., 2024).

**Limitation #2: The prompting schemes are not sufficiently optimized.** Existing schemes have untapped accuracy potential due to insufficient hyperparameter and prompt optimization. Pandey et al. (2025), for instance, transparently state that they do not assert optimality of their AGoT scheme and suggest that better prompts and hyperparameters may be discovered.

**Limitation #3: The prompting schemes are executed inefficiently.** All schemes rely on some form of extended test-time compute, i.e., generating more tokens during a response. However, many

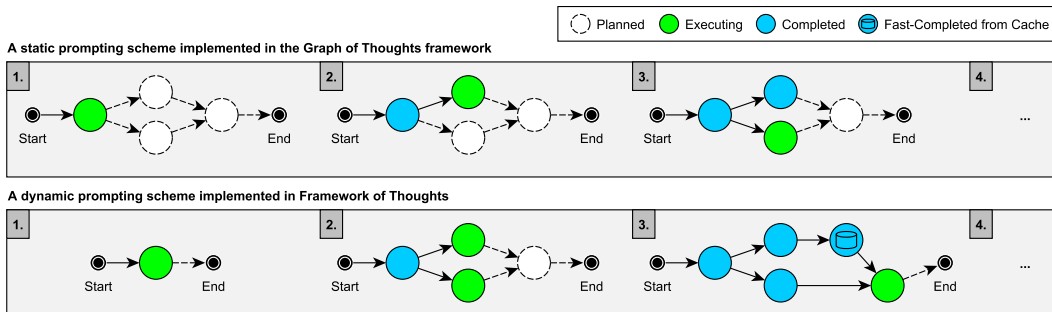

Figure 1: The execution graphs of a static prompting scheme implemented in the GoT framework versus a dynamic prompting scheme implemented in FoT. Nodes are operations, edges are information flows between them. While the graph structure is static and pre-planned in GoT, it can evolve dynamically in FoT (see steps 1-2). FoT executes operations in parallel (see step 2) and caches results of reoccurring operations (see step 3) to accelerate execution and reduce inference costs.

schemes are not time- and cost-efficient as they run LLM prompts sequentially and often perform the same LLM calls several times, thereby creating waiting times and unnecessary inference costs. In light of these widespread limitations of current schemes, we make two main contributions.

**Contribution #1: We introduce *Framework of Thoughts* (FoT)**, which is **not** a reasoning or prompting scheme itself but a foundation framework for implementing and optimizing reasoning schemes. Unlike previous frameworks such as the *Graph of Thoughts* (GoT) framework (Besta et al., 2024a), in which the prompting scheme of the same name was implemented, FoT comes with the following advantages (see Figure 1 for an illustrative comparison to the GoT framework):

(a) **Dynamic graph structures.** Unlike existing frameworks such as GoT or even *LangChain* (Inc., 2022) and *LangGraph* (Inc., 2024), which work well for static narrow-domain execution flows, FoT also enables prompting schemes that automatically and dynamically derive the graph structure, allowing the graph structure to change during execution.

(b) **Faster parallelized execution.** FoT executes operations concurrently whenever possible and introduces a set of dynamic execution constraints to protect the graph structure's logical integrity, i.e., to prevent race conditions while the graph structure evolves dynamically.

(c) **Cost savings through persistent caching.** FoT caches the results of all operations and re-uses cached results whenever possible, thereby preventing costly re-execution. Results can be cached temporarily within one execution/sample or persistently across multiple samples.

(d) **Optimized hyperparameters and prompts.** FoT has built-in tools for hyperparameter and prompt optimization, helping developers further optimize these often neglected factors. We show that substantial optimization really only becomes viable in combination with caching as the runtime and costs of the optimization procedure would otherwise be prohibitive.

FoT is designed as a modular and open framework, allowing users to specify any operations that can be defined in Python code, such as LLM calls, data retrieval, running code interpreters, or using other external tools. To readers unfamiliar with chain-, tree-, and graph-based prompting schemes, we strongly recommend viewing appendix Section A.1 and the corresponding Figure 5 for a concrete and detailed example of a prompting scheme implemented in FoT.

**Contribution #2: We empirically evaluate FoT's efficiency and optimization advantages.** We re-implement three popular prompting schemes, ToT, GoT, and *Probabilistic Tree-of-Thought Reasoning* (ProbTree) (Cao et al., 2023) in FoT to demonstrate the framework's universal applicability and possible optimization and efficiency gains. ProbTree is another suitable scheme for this demonstration, because it defines dynamic double-tree structures and also requires retrieval capabilities.

Table 1: Overview of popular prompting schemes, adapted from Besta et al. (2024b). We extended the table with the schemes in **bold**. "single" = single-prompt, "multi" = multi-prompt. "Dv." = derivation, "A" = automatic, "SA" = semi-automatic, "M" = manual. "R" = retrieval, "T" = tool use, "✓" = fully included, "(✓)" = partially included, "✗" = not included.

| Scheme | Citation | Topology | | | Pipeline | |
|--------|----------|----------|-------|-----|----------|-----|
| | | Class | Scope | Dv. | R | T |
| Chain of Thought (CoT) | (Wei et al., 2022) | chain | single | SA | ✗ | ✗ |
| Zero-Shot CoT | (Kojima et al., 2022) | chain | single | SA | ✗ | ✗ |
| Least-to-Most Prompting | (Zhou et al., 2023) | chain | multi | SA | ✗ | ✗ |
| Decomposed Prompting | (Khot et al., 2023) | chain | multi | SA | (✓) | (✓) |
| **Self-Discover** | (Zhou et al., 2024) | tree | single | SA | ✗ | ✗ |
| Self-Consistency CoT (CoT-SC) | (Wang et al., 2023) | tree | multi | SA | ✗ | ✗ |
| Tree of Thoughts (ToT) | (Yao et al., 2023) | tree | multi | SA | ✗ | ✗ |
| **Forest-of-Thought** | (Bi et al., 2024) | tree | multi | SA | (✓) | ✗ |
| Dynamic Least-to-Most Prompting | (Drozdov et al., 2023) | tree | multi | A | (✓) | ✗ |
| Skeleton-of-Thought (SoT) | (Ning et al., 2024) | tree | multi | A | ✗ | ✗ |
| Graph of Thoughts (GoT) | (Besta et al., 2024a) | graph | multi | M | ✗ | ✗ |
| Socratic Questioning (SQ) | (Qi et al., 2023) | graph | multi | SA | ✗ | ✗ |
| **Probabilistic ToT (ProbTree)** | (Cao et al., 2023) | graph | multi | SA | ✓ | ✗ |
| **Decompose-Analyze-Rethink** | (Xue et al., 2024) | graph | multi | SA | ✗ | ✗ |
| **Adaptive GoT (AGoT)** | (Pandey et al., 2025) | graph | multi | A | ✓ | ✗ |

## 2 OVERVIEW OF PROMPTING SCHEMES

Throughout this paper, we will use the terms *prompting schemes* and *reasoning schemes* interchangeably, as all schemes mentioned here incorporate both prompting and reasoning. A large number of prompting schemes have been proposed so far. Besta et al. (2024b) present a survey of the most relevant approaches along with a taxonomy to classify these schemes. The taxonomy distinguishes schemes by their **topology class** (chain, tree, or graph), **topology scope** (single-prompt or multi-prompt), **topology derivation** (automatic, semi-automatic, or manual), incorporation of different parts of the **generative AI pipeline** (e.g., retrieval or tool use), and other dimensions. Table 1 provides an overview of some of the most relevant prompting schemes according to this taxonomy. We added noteworthy schemes that were not yet identified by (Besta et al., 2024b). We note that all of these schemes could be implemented in FoT and thereby profit from the efficiency and optimization capabilities. To create awareness for the diversity of schemes and their specific requirements, we now summarize some key ideas of existing prompting schemes:

**Thinking step-by-step:** *Chain of Thought* (CoT) (Wei et al., 2022) includes examples of desired thought sequences (so-called *few-shot* examples) in the prompt, triggering the LLM to produce similar chains of thought when reasoning about a new problem. Kojima et al. (2022) show that adding a *"let's think step by step"* instruction to the prompt also elicits CoT responses from LLMs, known as *zero-shot* CoT. Both few-shot and zero-shot CoT are single-prompt schemes.

**Problem decomposition:** Some schemes such as *Decomposed Prompting* (Khot et al., 2023), *Least-to-Most Prompting* (Zhou et al., 2023), *Dynamic Least-to-Most Prompting* (Drozdov et al., 2023), and *Self-Discover* (Zhou et al., 2024) decompose complex problems into a series of subproblems, which are solved by subtask handlers. The decomposition is defined in the prompt (Decomposed Prompting), derived by the LLM ([Dynamic] Least-to-Most Prompting), or learned (Self-Discover).

**Question hierarchies:** *Socratic Questioning* (SQ) (Qi et al., 2023), *Probabilistic Tree-of-Thought Reasoning* (ProbTree) (Cao et al., 2023), and *Decompose-Analyze-Rethink* (DeAR) (Xue et al., 2024) recursively decompose complex questions into subquestions, thereby forming hierarchical question trees. They then answer the subquestions and use the answers to reason about the original question. ProbTree uses fact retrieval and dynamically chooses the most confident answer strategy.

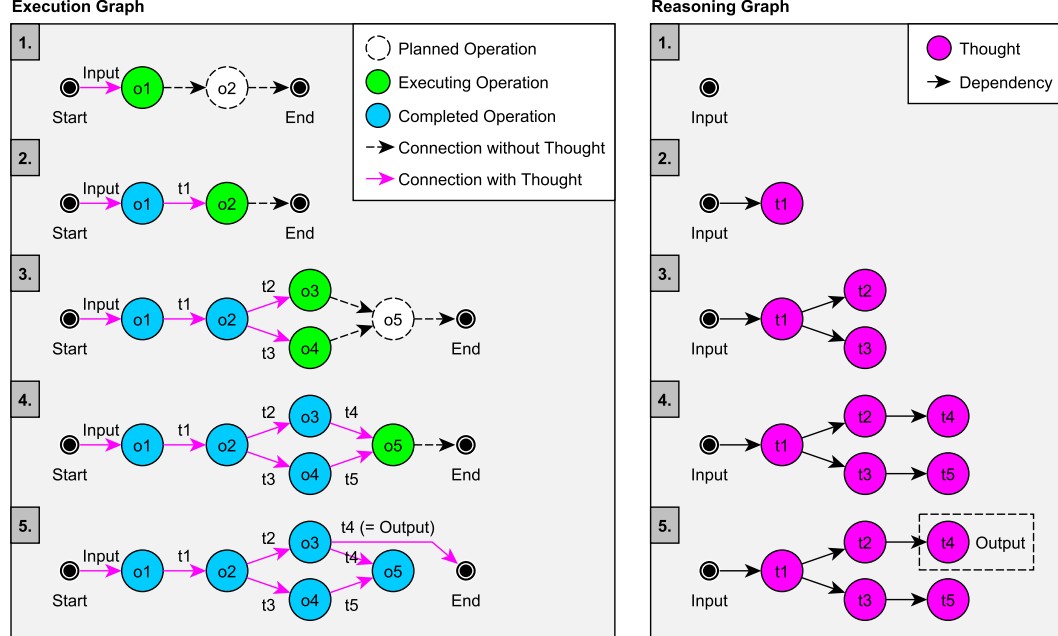

Figure 2: In the execution graph, nodes are operations and edges are connections that can carry thoughts. In the reasoning graph, nodes are thoughts and edges are dependencies. The execution graph can show the past, present, and future (i.e., completed, executing, and planned operations), whereas the reasoning graph only shows the past (thoughts produced by completed operations). The execution graph may be modified by operations, whereas the reasoning graph evolves as a byproduct.

**Trees and graphs:** *Tree of Thoughts* (ToT) (Yao et al., 2023) and *Graph of Thoughts* (GoT) (Besta et al., 2024a) arrange thoughts into task-specific tree and graph structures, respectively. They incorporate ideas such as exploration, backtracking, and iterative refinement. Both require users to define task-specific prompts and graph structures that define the task decomposition.

**Self-consistency:** *Self-Consistency* with CoT (CoT-SC) (Wang et al., 2023) and *Forest-of-Thought* (Bi et al., 2024) sample multiple answers to the same problem and select the most consistent one.

**Fully automatic:** *Skeleton-of-Thought* (SoT) (Ning et al., 2024) and *Adaptive Graph of Thought* (AGoT) (Pandey et al., 2025) are fully automatic reasoning schemes that do not require the user to define task-specific structures or prompts. They also incorporate parallel execution where possible. AGoT recursively calls itself if the LLM decides that further decomposition of a thought is required.

## 3 FRAMEWORK OF THOUGHTS

*Framework of Thoughts* (FoT) is a foundation framework for implementing and optimizing dynamic multi-prompt reasoning schemes based on chains, trees, or graphs. Unless stated otherwise, we will use the term *graph* to refer to all three of these structures.

### 3.1 DYNAMIC GRAPH STRUCTURES

In FoT, reasoning processes are modeled as one or more operations, whose inputs and outputs are chained together forming graphs. Operations can be anything that takes one or more input thoughts and returns one or more output thoughts, e.g., LLM calls, tool calls, code executions, or others. Thoughts can be any unit of information from a given universe of discourse $\mathcal{D}$. FoT distinguishes two types of graphs: The *execution graph* models *how* operations are executed and chained to arrive at an answer whereas the *reasoning graph* models *what* thoughts influence what other thoughts, making up the reasoning. See Figure 2 for an illustration and Figure 5 in the appendix for a concrete example.

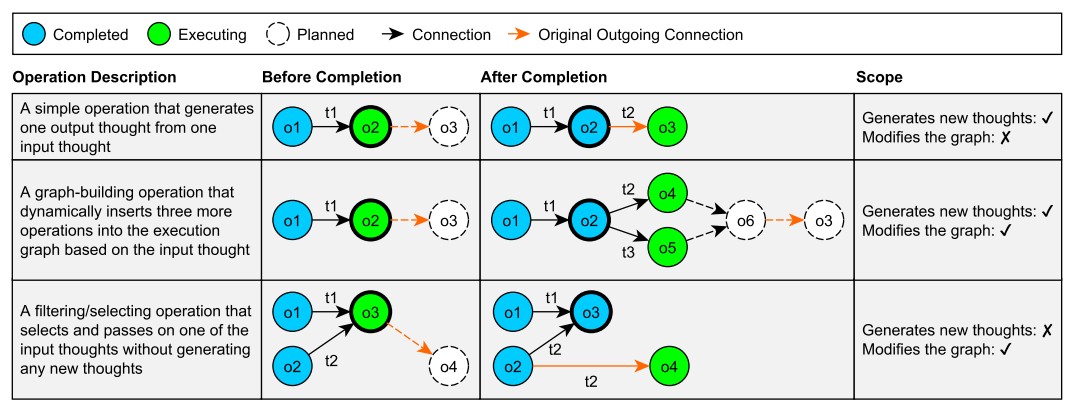

Figure 3: Non-exhaustive list with three example operations. Operations can generate new thoughts and/or modify the execution graph.

**Execution Graph:** The *execution graph* models the sequence of execution of the operations and how their inputs and outputs are connected. It retains the full history of *how* an answer came together and which operations were involved. At any given step $i$ during the execution, we define the execution graph as a directed multigraph

$$G_i^X = (O_i, E_i^X, s_i^X, t_i^X, \pi_i),$$

where:

- $O_i$ is the set of operations, each generating a set of output thoughts from a set of input thoughts,
- $E_i^X$ is the set of all connections between operations through which the operations receive and return thoughts,
- $s_i^X, t_i^X : E_i^X \to O_i$ map each connection to its source and target operations, respectively,
- $\pi_i : E_i^X \to \mathcal{D} \cup \{\bot\}$ maps each connection $e \in E_i^X$ to the thought $t \in \mathcal{D}$ that its source operation $s_i^X(e)$ generated for it, thereby keeping track of the current reasoning state at step $i$. If the source operation $s_i^X(e)$ has not yet been executed, then $\pi_i(e) = \bot$ (no thought).

We note that $\pi_i$ is similar to the *graph reasoning state* in GoT (Besta et al., 2024a) and the execution graph is similar to GoT's *graph of operations* (GoO). However, unlike the GoO, FoT's execution graph is not static but can be modified by the operations and evolve dynamically during execution, therefore the step index $i$.

**Reasoning Graph:** The *reasoning graph* is the result of the execution of operations and a simpler graph than the execution graph. It only describes *what* thoughts influenced what other thoughts but contains no information on which operations decided that these thoughts should be dependent. We define the reasoning graph as a directed graph:

$$G_i^R = (T_i, E_i^R, r_i),$$

where

- $T_i = \{\pi_i(e) \mid e \in E_i^X \land \pi_i(e) \neq \bot\}$ is the set of thoughts generated by the operations in the execution graph until step $i$,
- $E_i^R \subseteq T_i \times T_i$ is the set of dependencies between these thoughts (i.e., which though may have influenced which other thought),
- $r_i : T_i \to O_i$ maps each thought to the operation that generated it.

**Operations:** Operations are the building blocks of the execution graph. They perform the reasoning. An operation $o$ is a function that takes an execution graph and one or more input thoughts and returns an updated execution graph and one or more output thoughts:

Table 2: Definitions of ancestors, descendants, and exclusive descendants of an operation $o$.

| Set | Definition | Intuition |
|---|---|---|
| **Ancestors** | $A(o) = \{\, p \in O_i \mid \exists \text{ a directed path } p \rightsquigarrow o \,\}$ | Operations that $o$ depends on, directly or indirectly. |
| **Descendants** | $D(o) = \{\, d \in O_i \mid \exists \text{ a directed path } o \rightsquigarrow d \,\}$ | Operations that directly or indirectly depend on $o$'s output. |
| **Exclusive Descendants** | $E(o) = \{\, d \in D(o) \mid \forall l \in O_i \setminus \big(D(o) \cup \{o\}\big) : \forall \text{ directed paths } p : l \rightsquigarrow d, p \text{ goes through } o \}$ | Descendants that are only reachable via $o$. |

$$o : \mathcal{D}^* \times G^X \to \mathcal{D}^* \times G^X$$

where $\mathcal{D}^*$ is the set of all finite tuples of thoughts from the universe of discourse $\mathcal{D}$ and $G^X$ is the set of all possible execution graphs. This means that operations can do two things:

1. **Generate new thoughts:** Generate a set of output thoughts from a set of input thoughts.

2. **Modify the execution graph:** Modify the execution graph $G_i^X$ by adding or removing operations and connections yielding

$$G_{i+1}^X = (O_{i+1}, E_{i+1}^X, s_{i+1}^X, t_{i+1}^X, \pi_{i+1}),$$

where

$$O_{i+1} = (O_i \cup O^+) \setminus O^-,$$
$$E_{i+1}^X = (E_i^X \cup E^+) \setminus E^-,$$

$O^+$ and $E^+$ denote added operations and connections and $O^-$ and $E^-$ denote removed operations and connections. The projections $s_{i+1}^X$, $t_{i+1}^X$, $\pi_{i+1}$ are updated accordingly.

The latter ability of operations allows for automatically-derived dynamic graphs. This way, execution graphs may evolve while the operations making up the execution graph are being executed. See Figure 3 for three examples of operations.

**Initial execution graph:** While the execution graph can be automatically-derived by the operations, the user must specify at least the initial version of the execution graph $G_0^X$ containing at least one operation. This operation may then derive all subsequent operations and their connections dynamically based on the initial input.

## 3.2 SAFE PARALLEL EXECUTION

FoT's architecture includes a *Controller* and a *Scheduler* module. The Controller executes operations on the execution graph in an order defined by the Scheduler. It can run multiple operations **sequentially** or in **parallel**. The Scheduler only schedules operations that are ready to be executed. This generally means that all ancestor operations have been executed, meaning all input thoughts have been generated[1]. When operations are allowed to modify the execution graph, parallel execution poses a risk of race conditions, where conflicting or inconsistent graph modifications are attempted concurrently by multiple operations. To prevent non-deterministic outcomes and loss of information, we dynamically constrain the modifications that an operation $o$ is allowed to do (see Table 2 for a definition of relevant graph regions and Figure 4 for an illustration):

- $o$ can only see the subgraph induced by its ancestors $A(o)$, descendants $D(o)$, and itself.

- $o$ cannot modify the subgraph induced by its ancestors $A(o)$, as these operations have already been executed and the corresponding thoughts have been generated.

- $o$ cannot modify the subgraph induced by its non-exclusive descendants $D(o) \setminus E(o)$ as this might lead to race conditions with other parallel operations.

---

[1]An exception to this is an operation that only requires a subset of the input thoughts in order to execute.

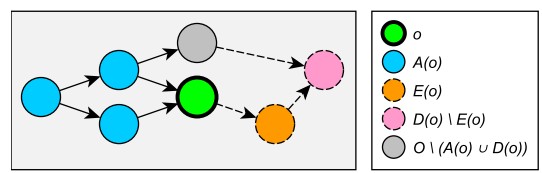

Figure 4: Illustration of graph regions defined in Table 2.

- $o$ can modify the subgraph induced by its exclusive descendants $E(o)$. It can add or remove operations and connections within this subgraph (including connections from $o$).

- $o$ can add new edges from ancestors $A(o)$ to exclusive descendants $E(o)$.

- $o$ can modify connections from exclusive descendants $E(o)$ or itself to descendants $D(o)$ by moving the start of these connections to any operation in $E(o) \cup A(o) \cup o$.

### 3.3 EFFICIENT CACHING

For many prompting schemes, the execution graph contains multiple instances of the same operation. Sometimes, these operations even execute with the same inputs. Instead of executing the potentially costly (e.g., LLM-based) operation again, FoT can cache and recall the previous operation outputs. FoT offers two types of caches: The **Process Cache** stores results temporarily within a single execution for a single problem instance. The **Persistent Cache** stores results persistently and can recall previous outputs even when executing subsequent problem instances from the dataset.

### 3.4 HYPERPARAMETER & PROMPT OPTIMIZATION

Almost all prompting schemes come with a set of prompts and several hyperparameters, such as those defining permissible graph structures, behavior of operations, or search and execution strategies. Since finding a set of well-performing prompts and hyperparameters is a non-trivial task, performance differences between prompting schemes may be due to suboptimal prompts and hyperparameters rather then architectural and methodological differences. To allow each prompting scheme to reach its potential, our FoT implementation includes a hyperparameter optimizer based on *Optuna* (Akiba et al., 2019) and a prompt optimizer based on *DSPy* (Khattab et al., 2023). We provide implementation details in Appendix A.3. FoT is open to various objective functions for optimization. This allows users to optimize their prompting schemes towards increased accuracy (typically against some validation set ground truth), decreased runtime, lower cost (e.g., based on prompt and response token count), or any (weighted) combination of these objectives.

## 4 EVALUATION

To evaluate the efficiency and optimization gains possible with FoT, we implement the three popular prompting schemes ToT, GoT, and ProbTree in FoT and apply them to five tasks that were used by the original authors of these schemes. We evaluate ToT on the *Game of 24* (Go24) (Yao et al., 2023), where the goal is to find an arithmetic expression that combines four numbers to reach 24. We evaluate GoT on *Sorting*, tasking the LLM to correctly sort a list of 128 integers, and *Document Merging* (DM), where the model must merge several documents into one; both problems were used in the original GoT paper (Besta et al., 2024a). We also implement ToT for Sorting. We evaluate ProbTree on *HotpotQA* (Yang et al., 2018) and *MuSiQue* (Trivedi et al., 2022), two multi-hop question-answering datasets. For all schemes and tasks, we then report the effect of adding parallelization and caching to the baseline implementation.

**Train-test split:** We split all datasets into training and test sets. The training sets are only used for optimization. All results reported in this paper are obtained on the test sets. For Go24, we use the same 100 test instances as Yao et al. (2023) and 200 different instances for training. Due to the small sizes of the Sorting and DM datasets, we split them equally into 50 train and 50 test instances. For HotpotQA and MuSiQue, we use 1,000 instances for training and 1,000 instances for test.

Table 3: Average runtime and cost per instance of all tested prompting schemes. "S" = Sequential execution. "P" = Parallel execution. "Process" = Process cache. "Persistent" = Persistent cache.

| | ToT | | GoT | | ProbTree | |
|---|---|---|---|---|---|---|
| | Go24 | Sorting | Sorting | DM | HotpotQA | MuSiQue |
| **Average runtime per instance**, in seconds (speed-up) | | | | | | |
| S+No cache | 782 (1.0x) | 452 (1.0x) | 259 (1.0x) | 145 (1.0x) | 12.8 (1.0x) | 21.6 (1.0x) |
| S+Process | 635 (1.2x) | 452 (1.0x) | 259 (1.0x) | 141 (1.0x) | 12.8 (1.0x) | 21.6 (1.0x) |
| S+Persistent | 373 (2.1x) | 452 (1.0x) | 259 (1.0x) | 140 (1.0x) | 12.8 (1.0x) | 18.4 (1.2x) |
| P+No cache | 31 (25.2x) | 39 (11.6x) | 30 (8.5x) | 32 (4.6x) | 6.8 (1.9x) | 10.4 (2.1x) |
| P+Process | 30 (25.8x) | 39 (11.6x) | 30 (8.5x) | 31 (4.7x) | 6.8 (1.9x) | 10.4 (2.1x) |
| P+Persistent | 22 (35.4x) | 39 (11.6x) | 30 (8.5x) | 31 (4.7x) | 6.8 (1.9x) | 8.9 (2.4x) |
| **Average cost per instance**, in USD cents (relative) | | | | | | |
| No cache | 29.6 (100%) | 15.9 (100%) | 5.0 (100%) | 6.9 (100%) | 0.5 (100%) | 0.8 (100%) |
| Process | 25.1 (85%) | 15.9 (100%) | 5.0 (100%) | 6.1 (88%) | 0.5 (100%) | 0.8 (100%) |
| Persistent | 16.1 (54%) | 15.9 (100%) | 5.0 (100%) | 5.9 (86%) | 0.5 (100%) | 0.7 (84%) |

**Metrics:** We measure performance on Go24 as the accuracy (percentage of correct answers; higher is better). Like Besta et al. (2024a), we score performance on Sorting by counting the number of mistakes (fewer mistakes are better). On DM, we calculate F1 scores from the redundancy and retention metrics (higher F1 scores are better). For HotpotQA and MuSiQue, we measure F1 scores (higher is better). We report runtime as the sum of the durations of all operations on the longest sequentially executed path and costs directly as the LLM API inference costs in USD.

**LLMs:** Yao et al. (2023) used a *GPT-4* model in their original ToT implementation. Due to budget restrictions, we instead use the cheaper *GPT-4o* on Go24. On Sorting and NDA, we use *GPT-3.5-Turbo*, as done by Besta et al. (2024a). This also makes the Sorting results of ToT and GoT comparable. The original ProbTree implementation by Cao et al. (2023) used an older *GPT-3* model, which is no longer available on OpenAI's API. We implement ProbTree with *GPT-4.1-mini* instead.

**Optimization:** We further optimize four schema implementations using FoT's optimization tools: We optimize the hyperparameters of ToT (for Go24 and Sorting), GoT (for Sorting), and the *Improve* prompt of GoT used in DM. As the optimization objective, we choose the respective task score (see metrics above) but constrain the costs to not exceed those of the unoptimized variant to prevent task score improvements stemming purely from more test-time compute.

## 4.1 RESULTS

Table 3 reports the efficiency gains (per-instance runtime and cost) possible for ToT, GoT, and Prob-Tree when implemented in FoT. Table 4 shows the task score improvements resulting from the optimization along with the total duration and cost of the optimization procedure. The per-instance runtime and cost of the optimized schemes can be found in Table 5 in the appendix.

It can be seen in Table 3 that FoT's parallelization and caching enable runtime accelerations between 1.9x and 35.4x on average, depending on the scheme and task. The average acceleration across all tasks (with parallel execution and persistent caching) is 10.7x, one order of magnitude faster than baseline implementations. While caching has no effect on Sorting and HotpotQA, it reduces the costs on Go24, DM, and MuSiQue by 14-46% on average. This shows that caching is not just effective on synthetic tasks such as Go24 but can also help on potential real-world tasks such as DM.

Table 4 shows that FoT's hyperparameter and prompt optimization tools could improve task scores while simultaneously reducing costs (see Table 5 in the appendix) on all evaluated schemes. While the optimization process itself incurs significant costs for exploring potential hyperparameter and prompt combinations, it can be seen that caching is particularly valuable during this procedure, re-

Table 4: Average task score improvements before and after optimization as well as total duration and cost of optimization. Abbreviations as in Table 3. Measured scores are accuracy (Go24), mistakes (Sorting), F1 score (DM). "↑" = Higher score is better. "↓" = Lower score is better.

| | ToT | | GoT | |
|---|---|---|---|---|
| **Score** | Go24 ↑ | Sorting ↓ | Sorting ↓ | DM ↑ |
| Original | 63.0% | 18.4 | 12.7 | 8.4 |
| Optimized | **66.0%** | **18.2** | **12.1** | **8.8** |
| **Total optimization duration**, in minutes (speed-up) | | | | |
| S+No cache | 39,596 (1.0x) | 14,372 (1.0x) | 8,371 (1.0x) | 3,838 (1.0x) |
| S+Process | 30,576 (1.3x) | 14,372 (1.0x) | 8,371 (1.0x) | 3,804 (1.0x) |
| S+Persistent | 9,294 (4.3x) | 2,136 (6.7x) | 822 (10.2x) | 1,638 (2.3x) |
| P+No cache | 2,603 (15.2x) | 1,928 (7.5x) | 1,195 (7.0x) | 1,225 (3.1x) |
| P+Process | 2,548 (15.5x) | 1,928 (7.5x) | 1,195 (7.0x) | 1,216 (3.2x) |
| P+Persistent | 788 (50.2x) | 418 (34.4x) | 196 (42.6x) | 441 (8.7x) |
| **Total optimization cost**, in USD (relative) | | | | |
| No cache | 1,224.41 (100%) | 303.18 (100%) | 135.74 (100%) | 99.35 (100%) |
| Process | 917.61 (75%) | 303.18 (100%) | 135.74 (100%) | 95.17 (96%) |
| Persistent | 153.56 (13%) | 46.10 (15%) | 12.27 (9%) | 36.15 (36%) |

ducing optimization costs to only 9-36% of the costs that would have been incurred without caching. Similarly, FoT accelerated the total duration of the optimization procedure by a factor of up to 8.7-50.2. This highlights the fact that optimization of reasoning schemes often only becomes viable when parallelization and caching are used, as the required time and budget could otherwise be prohibitive.

## 5 CONCLUSION

We introduced Framework of Thoughts (FoT), a foundation framework for implementing and optimizing prompting schemes. Unlike previous frameworks such as the Graph of Thoughts framework, FoT can not only model static graph structures but also dynamic graph structures that evolve during execution. This paves the way for a new generation of adaptive prompting schemes that can reason effectively, also about previously unseen or highly heterogeneous problem types. Schemes implemented in FoT benefit from FoT's "efficient-by-default" setup that accelerates runtimes through parallelization and saves inference costs through extensive caching. We empirically show how this can accelerate existing schemes by an order of magnitude and cut cost nearly in half in some cases. Lastly, FoT provides developers with tools to optimize hyperparameters and prompts of their prompting schemes. Using these tools, we identified configurations with better accuracy and simultaneously lower costs for the popular schemes Tree of Thoughts and Graph of Thoughts.

**Limitations & Future Work**   We implemented one manual (GoT) and two semi-automatic (ToT and ProbTree) prompting schemes in FoT. While GoT's execution graph is entirely static, ToT and ProbTree exhibit at least some degree of the dynamic graph modifications that FoT was designed to model. Still, we encourage others to implement new fully-automatic prompting schemes in FoT that demonstrate the advantage of dynamic graphs on more problem types than this paper did. For FoT itself, we envision future improvements such as parallel optimization of multiple prompts on different prompt optimization techniques such as *GEPA* (Agrawal et al., 2025), joint optimization of hyperparameters and prompts, more user-friendly graph abstractions and interfaces, as well as more relaxed graph modification rules during parallel execution.

**Reproducibility Statement**    To ensure reproducibility of our results and enable others to build new prompting schemes in FoT, we share our entire codebase[2]. Our exact prompting scheme implementations and the FoT code can be found there. The repository further includes instructions on how to install the framework and run the dataset evaluations and optimization studies, a simple test example in a *Jupyter* notebook to familiarize oneself with the framework, as well as dataset evaluation and optimization outputs. Readers looking to re-implement parts of our code can also find further implementation details in appendix Sections A.2 (schema explanations) and A.7 (schema prompts), as well as Table 6 (schema hyperparameters).

**Ethics Statement**    We apply different prompting schemes to tasks such as merging documents or answering complex questions. These tasks can hold significant value in some domains and situations. However, we stress that prompting schemes based on LLMs are not perfect and can result in incorrect yet plausible answers. Users of such prompting schemes, whether implemented in FoT or not, should exhibit caution and always cross-check outputs for correctness, especially in high-stakes scenarios.

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

# A   APPENDIX

## A.1   ILLUSTRATIVE EXAMPLE

Figure 5 provides a concrete example of a *Tree of Thoughts* (ToT) prompting scheme implemented in *Framework of Thoughts* (FoT). The figure shows the step-by-step evolution of a dynamic execution graph on the *Game of 24* task. In the Game of 24, the goal is to combine four given numbers using arithmetic operations ($+$, $-$, $\times$, $\div$) and brackets to reach 24. The shown implementation takes the four numbers $[1, 2, 3, 4]$ as input and explores possible arithmetic combinations of these numbers until it finds and returns a full arithmetic expression $(4 \times (2 \times 3)) \div 1 = 24$.

The actual ToT implementation for Game of 24 used in this paper is more complex than the implementation shown in Figure 5 to stay true to the original ToT implementation by Yao et al. (2023), which proposes more than three new thoughts in each *Propose* prompt and samples multiple *Value* prompt estimations for each thought chain, to name just two differences. See Section A.2 for a description of our implementation.

## A.2   DETAILED EXPLANATION OF THE METHODS

In the following, a detailed explanation of each method is given. Variables in `monospace` are hyperparameters that can be optimized.

**ToT on *Game of 24* (Go24)**   The goal of *Game of 24* is to form a mathematical expression with four given numbers using $+$, $-$, $\times$, $\div$, and brackets that equals 24. One example for the input numbers $[1, 2, 3, 4]$ would be $(4 \times (2 \times 3)) \div 1 = 24$.

The ToT implementation uses an iterative process as follows:

A *Propose* operation creates `number of examples` many expressions that could lead in the right direction, as well as the remaining numbers. As an example, for $[1, 2, 3, 4]$ it could output the expressions $1 + 2 = 3$ with the remainder $[3, 4, 3]$, or $2 \times 3 = 6$ with the remainder $[1, 4, 6]$.

Each of these proposals are then scored by an LLM (*Value* operation) `number of samples` times into "SURE", "LIKELY", or "IMPOSSIBLE" to reach 24 at some point. The scores for each proposal are turned into floating point numbers and a *Filter* operation keeps only the `keep top N` candidates.

The next *Propose* operation then creates new expressions and remainders for the remainders of the top candidates, and the process repeats until only one number is left.

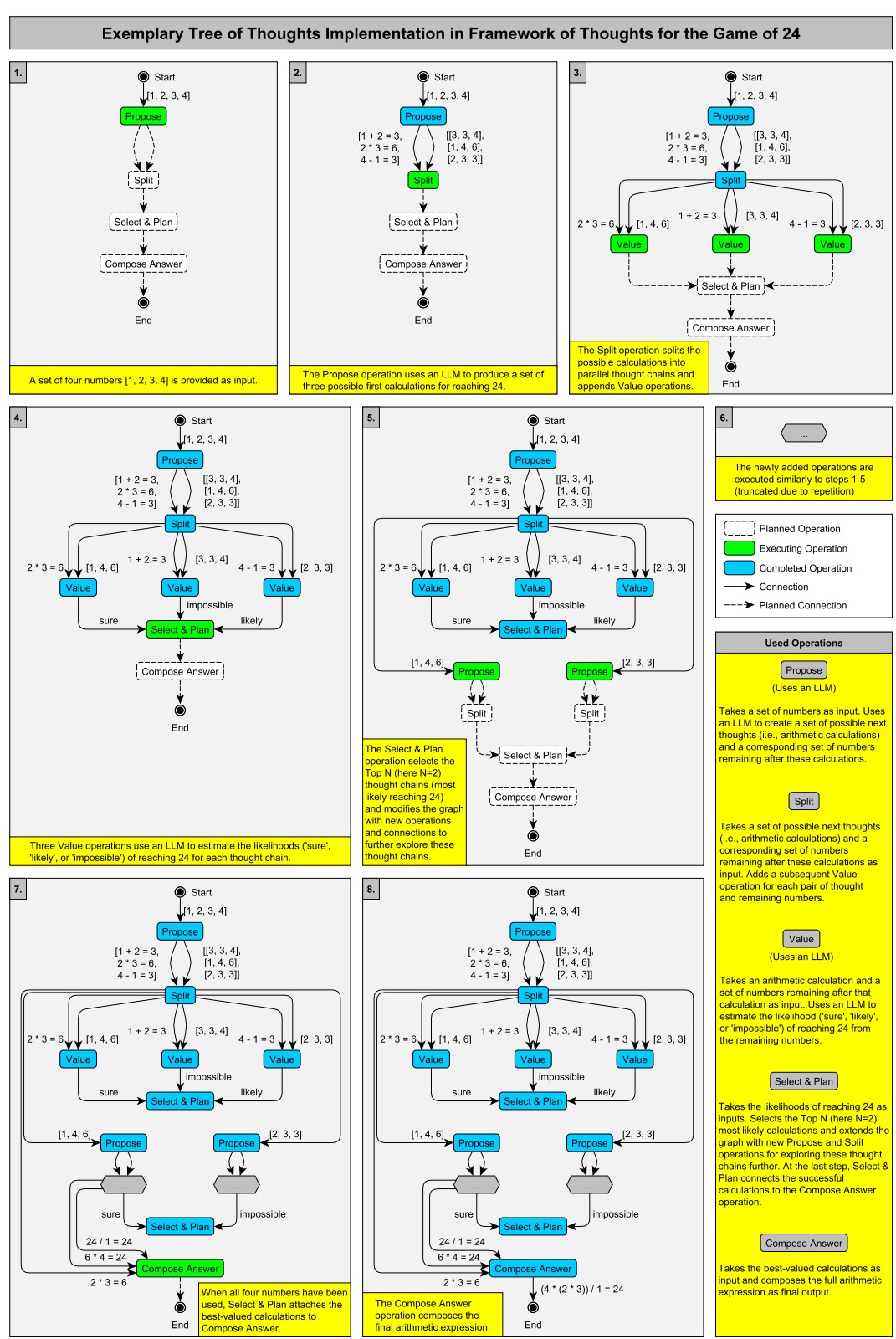

Figure 5: Exemplary evolution of the execution graph in an implementation of the *Tree of Thoughts* prompting scheme in *Framework of Thoughts* for the *Game of 24* (see explanation in Section A.1).

**ToT and GoT on *Sorting*** The goal of *Sorting* is to sort an array of 128 digits into ascending order with repetition. An example on the 8 digit input $[1, 6, 4, 0, 3, 4, 6, 7]$ would be $[0, 1, 3, 4, 4, 6, 6, 7]$.

The ToT implementation uses an iterative process as follows:

`number of branches` LLM operations create a first sorted candidate. Each of these candidates is evaluated to how many mistakes have been made and the top candidate is kept.

`number of branches` LLM operations then try to figure out the mistakes and improve on the candidates. This process is repeated `improvement levels` times, after which the best candidate is returned.

The GoT implementation uses the following divide-and-conquer approach:

The list is split into 8 short lists. Each of them gets sorted by an LLM operation `number of sort branches` times in parallel and the best scored one of each is kept. After that, two neighbouring lists are merged by an LLM operation `number of merge branches` times in parallel and the best scored one is kept. This is repeated twice to end up with one sorted list of 128 digits.

After that, an LLM operation repairs the list in sequence `global improvement rounds` before returning.

**GoT on *Document Merging* (DM)** The goal of this process is to merge 4 documents (here Non-Disclosure Agreements, NDAs) into a single document, hereby maximizing retaining information and minimizing redundancy.

The GoT implementation uses the following iterative approach:

First, `number of merges` LLM operations merge all NDAs into one. An LLM scores the redundancy and retention in the mergers between 0 and 10, of which the harmonic mean is calculated.

The top `keep best merges` candidates are then aggregated `number of aggregations` times using an LLM operation to an improved candidate, which is then also scored.

Of all candidates from both stages, the best is then improved using an LLM operation `number of improvements` times before being returned.

The generative LLM calls are executed at temperature 1 while the scoring is performed at temperature 0.

**ProbTree on *HotpotQA* and *MuSiQue*** The goal of both *HotpotQA* and *MuSiQue* is to answer a multi-hop question such as "ARE BOTH SUPERDRAG AND COLLECTIVE SOUL ROCK BANDS?", with *MuSiQue* being more challenging than *HotpotQA*.

The Probtree implementation uses the following process:

At first, an LLM operation using few-shot prompting generates a tree structure subdividing the original question into a tree of questions that are separately answerable. In the example above, these could be "IS SUPERDRAG A ROCK BAND?" and "IS COLLECTIVE SOUL A ROCK BAND?".

Each of the leaf nodes are then answered by an LLM operation closed book (meaning no supportive evidence is provided), and open book (a retriever adds supportive information using *BM25* on a Wikipedia dump).

After that, the tree is traversed upwards and each parent question is answered open book, closed book, and based on the aggregated evidence from their child nodes.

Each LLM operation emits token-level log likelihoods that are used to filter the best answer with the highest confidence.

For detailed intricacies of the tree structure, how dependencies between questions can be mapped, and the process of filtering based on token-level log likelihoods, see the original paper (Cao et al., 2023).

### A.3 Hyperparameter & Prompt Optimization

**Hyperparameter Optimization**    A hyperparameter can be any variable that

- models variations of prompts and parsers in an LLM-based operation, (e.g., chooses from a set of possible prompts),

- influences how graph-modifying operations change the graph structure, (e.g., what operations to perform next based on the inputs),

- influences the inital execution graph structure, (e.g., number of branches in a ToT prompting scheme),

- influences the Scheduler's search strategy, (e.g., breadth-first, depth-first), or

- sets the number of permissible concurrent executions in the Controller.

For hyperparameter optimization, our FoT implementation uses *Optuna* (Akiba et al., 2019), which includes a variety of hyperparameter optimization techniques. We use a *tree-structured Parzen estimator* (TPE) (Watanabe, 2023) which is a *sequential model-based optimization* (SMBO) technique. Hyperparameters may be conditional and can be categorical, discrete or continuous. TPE models the search space by estimating two probability densities: one over the best-performing hyperparameters and one over the others. It applies *kernel density estimators* to model these distributions. New candidates are proposed by maximizing the expected improvement, which reduces to maximizing the ratio of the two densities.

**Prompt Optimization**    To optimize the formulation of prompts, our FoT implementation integrates *Cooperative Prompt Optimization* (COPRO) via *DSPy* (Khattab et al., 2023). Starting with an initial prompt formulation, COPRO applies an evolutionary algorithm to generate offspring variations of the instruction part of the prompt, selecting the best-performing variations, and generating new variations of these again.

**Optimization Objective**    The optimization objective can be either based on the ground truth for a given test set (e.g., an accuracy measure), the output of the execution, and/or measurements that are computed during execution such as prompt and response token count or cost, a predefined execution cost per operation, or execution latency. The objective can also be any (weighted) combination of the above. As the objective function may be a noisy non-differentiable blackbox function, we cannot use gradient-based optimizers. Instead, we rely on hyperparameter optimizers that can efficiently explore the hyperparameter space.

### A.4 Evaluations for Probtree

For the retrieval step in Probtree, we use BM25 on the October 2017 Wikipedia dump, the same dataset as used in the original Probtree paper (Cao et al., 2023). However, we use the same dataset for retrieval for *MuSiQue* unlike the original implementation.

The resulting F1 score for the dataset evaluation of Probtree on *HotpotQA* is $53.8\%$ and on *MuSiQue* is $24.7\%$.

### A.5 Optimization Experiments

Table 5 shows the per-instance runtime and cost of our optimized ToT, and GoT variants.

For both Sorting tasks, 50 iterations were run. On the Game of 24 task, 25 iterations were run due to cost restrictions. Original and optimized hyperparameters as well as the acceptable parameter range during optimization can be found in Table 6.

The parameters used in the COPRO prompt optimization on the Document Merging task are: depth: 6, keep top: 8, breadth: 8. The prompt proposal language model is *GPT-4o-mini* at a temperature of 1.6.

Table 5: Average runtime and cost per instance for optimized schemes. Abbreviations as in Table 3.

| | ToT | | GoT | |
|---|---|---|---|---|
| | Go24 | Sorting | Sorting | DM |
| **Optimized scheme: Average runtime per instance**, in seconds (in % of unoptimized scheme) | | | | |
| S+No cache | 452 (58%) | 398 (88%) | 239 (92%) | 101 (70%) |
| S+Process | 346 (55%) | 398 (88%) | 239 (92%) | 100 (71%) |
| S+Persistent | 195 (52%) | 398 (88%) | 239 (92%) | 100 (71%) |
| P+No cache | 27 (86%) | 47 (119%) | 29 (95%) | 27 (86%) |
| P+Process | 26 (86%) | 47 (119%) | 29 (95%) | 27 (86%) |
| P+Persistent | 21 (96%) | 47 (119%) | 29 (95%) | 27 (86%) |
| **Optimized scheme: Average cost per instance**, in USD cents (in % of unoptimized scheme) | | | | |
| No cache | 25.2 (85%) | 15.8 (99%) | 4.9 (98%) | 5.9 (87%) |
| Process | 20.1 (80%) | 15.8 (99%) | 4.9 (98%) | 5.8 (95%) |
| Persistent | 12.3 (76%) | 15.8 (99%) | 4.9 (98%) | 5.8 (97%) |

Table 6: Original and optimized hyperparameters alongside the ranges used in the optimization.

| Hyperparameter | Original | Optimized | Range |
|---|---|---|---|
| **Task: Game of 24 (ToT)** | | | |
| number of examples | 8 | 11 | $[4, 12]$ |
| samples | $(3, 3, 3)$ | $(3,2,2)$ | $[1, 5]$ each |
| keep top N (layers 1, 2) | $(5, 5)$ | $(5, 3)$ | $[2, min(7, |input|)]$ each |
| **Task: Sorting (ToT)** | | | |
| number of branches | 20 | 14 | $[5, 20]$ |
| improvement levels | 4 | 6 | $[1, 6]$ |
| **Task: Sorting (GoT)** | | | |
| number of sort branches | 5 | 2 | $[1, 10]$ |
| number of merge branches | 10 | 13 | $[5, 25]$ |
| global improvement rounds | 1 | 2 | $[1, 3]$ |

## A.6 OPTIMIZED PROMPT

The original unoptimized *Improve* prompt can be seen in Listing 1 and the optimized *Improve* prompt in Listing 2.

Listing 1: Original Improve prompt only with USER message.

```
USER:
The following NDA  merges initial NDAs <Doc1> - <DocN>.
Please improve the summary NDA  by adding more information
and removing redundancy. Output only the improved NDA, placed
between the two tags <Merged> and </Merged>, without any
additional text.

Here are NDAs <Doc1> - <DocN>:

<Doc1>
[Content of NDA 1]
</Doc1>

<Doc2>
[Content of NDA 2]
</Doc2>
```

```
...

<DocN>
[Content of NDA N]
</DocN>

Here is the summary NDA :

[Content of summary]

```

Listing 2: Optimized Improve prompt with SYSTEM and USER message.

```
SYSTEM:
Your input fields are:
1. 'summaries' (list[str]): The summaries of the NDAs to improve
2. 'docs' (list[str]): The original NDAs
Your output fields are:
1. 'merged' (str): The improved summary of the NDAs
All interactions will be structured in the following way, with the
    appropriate values filled in.

[[ ## summaries ## ]]
{summaries}

[[ ## docs ## ]]
{docs}

[[ ## merged ## ]]
{merged}

[[ ## completed ## ]]
In adhering to this structure, your objective is:
Generate a succinct, informative, and harmonized summary of the provided
non-disclosure agreements (NDAs) by meticulously blending insights from
    both
the accompanying summaries and the original documents. Your output should
encapsulate critical details while enhancing clarity and legibility,
    minimizing
any excessive language. Highlight and comport essential legal terms
    appropriately,
ensuring the intent and key clauses remain conspicuous. Finalize your
    summary
formatted within the designated <Merged> and </Merged> tags aimed at
    promoting
swift understanding and seamless usage of the key information presented.
Ensure to format the output between <Merged> and </Merged> tags.

USER:
[[ ## summaries ## ]]
[Content of summary]

[[ ## docs ## ]]
[Content of NDAs]

Respond with the corresponding output fields, starting with:
[[ ## merged ## ]]
and end with:
[[ ## completed ## ]]
```

A.7 PROMPTS USED IN THE PAPER

**ToT on *Game of 24* (Go24)**   The prompts for this task were taken from the original implementation of Game of 24 (Yao et al., 2023) with additional system messages to ensure newer models follow the exact few-shot prompt. The prompts used are *Propose* (Listing 3), *Value* (Listing 4), and *LastStepValue* (Listing 5). The system message in the *Propose* prompt also contains a parameter to control the number of examples.

Listing 3: Propose prompt for Game of 24 with SYSTEM and USER message.

```
SYSTEM:
Follow exactly the few shot prompt. Output exactly [num_examples] next
    steps.

USER:
Input: 2 8 8 14
Possible next steps:
2 + 8 = 10 (left: 8 10 14)
8 / 2 = 4 (left: 4 8 14)
14 + 2 = 16 (left: 8 8 16)
2 * 8 = 16 (left: 8 14 16)
8 - 2 = 6 (left: 6 8 14)
14 - 8 = 6 (left: 2 6 8)
14 /  2 = 7 (left: 7 8 8)
14 - 2 = 12 (left: 8 8 12)
Input: [input_list]
Possible next steps:
```

Listing 4: Value prompt for Game of 24 with SYSTEM and USER message.

```
SYSTEM:
Follow exactly the few shot prompt.

USER:
Evaluate if given numbers can reach 24 (sure/likely/impossible)
10 14
10 + 14 = 24
sure
11 12
11 + 12 = 23
12 - 11 = 1
11 * 12 = 132
11 / 12 = 0.91
impossible
4 4 10
4 + 4 + 10 = 8 + 10 = 18
4 * 10 - 4 = 40 - 4 = 36
(10 - 4) * 4 = 6 * 4 = 24
sure
4 9 11
9 + 11 + 4 = 20 + 4 = 24
sure
5 7 8
5 + 7 + 8 = 12 + 8 = 20
(8 - 5) * 7 = 3 * 7 = 21
I cannot obtain 24 now, but numbers are within a reasonable range
likely
5 6 6
5 + 6 + 6 = 17
(6 - 5) * 6 = 1 * 6 = 6
I cannot obtain 24 now, but numbers are within a reasonable range
likely
10 10 11
10 + 10 + 11 = 31
(11 - 10) * 10 = 10
```

```
10 10 10 are all too big
impossible
1 3 3
1 * 3 * 3 = 9
(1 + 3) * 3 = 12
1 3 3 are all too small
impossible
[left]
```

Listing 5: LastStepValue prompt for Game of 24 with SYSTEM and USER message.

```
SYSTEM:
Follow exactly the few shot prompt.

USER:
Use numbers and basic arithmetic operations (+ - * /) to obtain 24. Given
    an input and an answer, give a judgement (sure/impossible) if the
    answer is correct, i.e. it uses each input exactly once and no other
    numbers, and reach 24.
Input: 4 4 6 8
Answer: (4 + 8) * (6 - 4) = 24
Judge:
sure
Input: 2 9 10 12
Answer: 2 * 12 * (10 - 9) = 24
Judge:
sure
Input: 4 9 10 13
Answer: (13 - 9) * (10 - 4) = 24
Judge:
sure
Input: 4 4 6 8
Answer: (4 + 8) * (6 - 4) + 1 = 25
Judge:
impossible
Input: 2 9 10 12
Answer: 2 * (12 - 10) = 24
Judge:
impossible
Input: 4 9 10 13
Answer: (13 - 4) * (10 - 9) = 24
Judge:
impossible
Input: [left]
Answer: [answer]
```

**ToT and GoT on** *Sorting* The prompts for this task were taken from the original implementation of Sorting (Besta et al., 2024a), with minor adjustments. The only contain user messages. The prompts used are *Generate* (for both ToT and GoT) (Listing 6), *Improve* (ToT) (Listing 7), *Split* (GoT) (Listing 8), and *Aggregate* (GoT) (Listing 9).

Listing 6: Generate prompt for Sorting.

```
USER:
<Instruction> Sort the following list of numbers in ascending order.
    Output only the sorted list of numbers, no additional text. </
    Instruction>

<Examples>
Input: [5, 1, 0, 1, 2, 0, 4, 8, 1, 9, 5, 1, 3, 3, 9, 7]
Output: [0, 0, 1, 1, 1, 1, 2, 3, 3, 4, 5, 5, 7, 8, 9, 9]

Input: [3, 7, 0, 2, 8, 1, 2, 2, 2, 4, 7, 8, 5, 5, 3, 9, 4, 3, 5, 6, 6, 4,
    4, 5, 2, 0, 9, 3, 3, 9, 2, 1]
```

```
1026  Output: [0, 0, 1, 1, 2, 2, 2, 2, 2, 2, 3, 3, 3, 3, 3, 4, 4, 4, 4, 5, 5,
1027      5, 5, 6, 6, 7, 7, 8, 8, 9, 9, 9]
1028
1029  Input: [4, 4, 9, 7, 9, 7, 0, 0, 4, 9, 1, 7, 9, 5, 8, 7, 5, 6, 3, 8, 6, 7,
1030      5, 8, 5, 0, 6, 3, 7, 0, 5, 3, 7, 5, 2, 4, 4, 9, 0, 7, 8, 2, 7, 7, 7,
1031      2, 1, 3, 9, 9, 7, 9, 6, 6, 4, 5, 4, 2, 0, 8, 9, 0, 2, 2]
1032  Output: [0, 0, 0, 0, 0, 0, 0, 1, 1, 2, 2, 2, 2, 2, 2, 3, 3, 3, 3, 4, 4,
1033      4, 4, 4, 4, 5, 5, 5, 5, 5, 5, 6, 6, 6, 6, 6, 7, 7, 7, 7, 7, 7,
1034      7, 7, 7, 7, 7, 7, 8, 8, 8, 8, 8, 9, 9, 9, 9, 9, 9, 9, 9, 9]
1034  </Examples>
1035
1036  Input: [input_list]
```

Listing 7: ToT Improve prompt for Sorting.

```
1039  USER:
1040  <Instruction> The following two lists represent an unsorted list of
1041      numbers and a sorted variant of that list. The sorted variant is not
1042      correct. Fix the sorted variant so that it is correct.
1043  Make sure that the output list is sorted in ascending order, has the same
1044      number of elements as the input list ([length of input_list]), and
1044      contains the same elements as the input list. </Instruction>
1045
1046  <Approach>
1047  To fix the incorrectly sorted list follow these steps:
1047  1. For each number from 0 to 9, compare the frequency of that number in
1048      the incorrectly sorted list to the frequency of that number in the
1049      input list.
1050  2. Iterate through the incorrectly sorted list and add or remove numbers
1051      as needed to make the frequency of each number in the incorrectly
1052      sorted list match the frequency of that number in the input list.
1052  </Approach>
1053
1054  <Examples>
1055  Input: [3, 7, 0, 2, 8, 1, 2, 2, 2, 4, 7, 8, 5, 5, 3, 9]
1056  Incorrectly Sorted: [0, 0, 0, 0, 0, 1, 2, 2, 3, 3, 4, 4, 4, 5, 5, 7, 7,
1056      8, 8, 9, 9, 9, 9]
1057  Reason: The incorrectly sorted list contains four extra 0s, two extra 4s
1058      and three extra 9s and is missing two 2s.
1059  Output: [0, 1, 2, 2, 2, 2, 3, 3, 4, 5, 5, 7, 7, 8, 8, 9]
1060
1061  Input: [6, 4, 5, 7, 5, 6, 9, 7, 6, 9, 4, 6, 9, 8, 1, 9, 2, 4, 9, 0, 7, 6,
1061      5, 6, 6, 2, 8, 3, 9, 5, 6, 1]
1062  Incorrectly Sorted: [0, 1, 1, 2, 2, 3, 4, 4, 4, 4, 4, 5, 5, 5, 5, 6, 6,
1063      6, 6, 6, 6, 7, 7, 7, 8, 8, 9, 9, 9, 9, 9]
1064  Reason: The incorrectly sorted list contains two extra 4s and is missing
1065      two 6s and one 9.
1066  Output: [0, 1, 1, 2, 2, 3, 4, 4, 4, 5, 5, 5, 5, 6, 6, 6, 6, 6, 6, 6, 6,
1067      7, 7, 7, 8, 8, 9, 9, 9, 9, 9, 9]
1068
1068  Input: [4, 4, 9, 7, 9, 7, 0, 0, 4, 9, 1, 7, 9, 5, 8, 7, 5, 6, 3, 8, 6, 7,
1069      5, 8, 5, 0, 6, 3, 7, 0, 5, 3, 7, 5, 2, 4, 4, 9, 0, 7, 8, 2, 7, 7, 7,
1070      2, 1, 3, 9, 9, 7, 9, 6, 6, 4, 5, 4, 2, 0, 8, 9, 0, 2, 2]
1071  Incorrectly Sorted: [0, 0, 0, 0, 0, 0, 0, 1, 1, 2, 2, 2, 2, 3, 3, 3, 4,
1072      4, 4, 4, 5, 5, 5, 5, 5, 6, 6, 6, 6, 7, 7, 7, 7, 7, 7, 8, 8, 8, 8, 8,
1073      8, 9, 9, 9, 9, 9, 9, 9, 9]
1073  Reason: The incorrectly sorted list contains one extra 8 and is missing
1074      two 2s, one 3, three 4s, two 5s, one 6, six 7s and one 9.
1075  Output: [0, 0, 0, 0, 0, 0, 0, 1, 1, 2, 2, 2, 2, 2, 2, 3, 3, 3, 3, 4, 4,
1076      4, 4, 4, 4, 4, 5, 5, 5, 5, 5, 5, 5, 6, 6, 6, 6, 6, 7, 7, 7, 7, 7, 7,
1077      7, 7, 7, 7, 7, 7, 8, 8, 8, 8, 8, 9, 9, 9, 9, 9, 9, 9, 9, 9]
1077  </Examples>
1078
1079  Input: [input_list]
1079  Incorrectly Sorted: [incorrectly_sorted]
```

Listing 8: GoT Split prompt for Sorting.

```
USER:
<Instruction> Split the following list of 128 numbers into 8 lists of 16
    numbers each, the first list should contain the first 16 numbers, the
     second list the second 16 numbers, the third list the third 16
    numbers, the fourth list the fourth 16 numbers, the fifth list the
    fifth 16 numbers and so on.
Only output the final 8 lists in the following format without any
    additional text or thoughts!:
{
    "List 1": [3, 4, 3, 5, 7, 8, 1, ...],
    "List 2": [2, 9, 2, 4, 7, 1, 5, ...],
    "List 3": [6, 9, 8, 1, 9, 2, 4, ...],
    "List 4": [9, 0, 7, 6, 5, 6, 6, ...],
    "List 5": [7, 9, 4, 1, 1, 8, 1, ...],
    "List 6": [1, 9, 0, 4, 3, 3, 5, ...],
    "List 7": [2, 4, 3, 5, 8, 2, 2, ...],
    "List 8": [4, 2, 1, 2, 7, 6, 8, ...]
} </Instruction>

<Example>
Input: [6, 0, 2, 3, 8, 3, 0, 2, 4, 5, 4, 1, 3, 6, 9, 8, 3, 1, 2, 6, 5, 3,
    9, 8, 9, 1, 6, 1, 0, 2, 8, 9, 5, 3, 1, 2, 7, 9, 4, 8, 8, 9, 3, 2, 8,
    4, 7, 4, 3, 8, 7, 3, 6, 4, 0, 0, 6, 8, 1, 5, 8, 7, 5, 1, 4, 0, 8, 6,
    1, 3, 6, 1, 7, 6, 8, 7, 3, 7, 8, 2, 0, 8, 2, 6, 0, 0, 9, 9, 8, 6, 9,
    4, 8, 5, 5, 0, 0, 9, 3, 9, 4, 0, 5, 6, 2, 4, 6, 7, 7, 7, 8, 0, 4, 9,
    1, 4, 8, 5, 1, 4, 4, 7, 4, 9, 3, 9, 6, 7]
Output:
{
    "List 1": [6, 0, 2, 3, 8, 3, 0, 2, 4, 5, 4, 1, 3, 6, 9, 8],
    "List 2": [3, 1, 2, 6, 5, 3, 9, 8, 9, 1, 6, 1, 0, 2, 8, 9],
    "List 3": [5, 3, 1, 2, 7, 9, 4, 8, 8, 9, 3, 2, 8, 4, 7, 4],
    "List 4": [3, 8, 7, 3, 6, 4, 0, 0, 6, 8, 1, 5, 8, 7, 5, 1],
    "List 5": [4, 0, 8, 6, 1, 3, 6, 1, 7, 6, 8, 7, 3, 7, 8, 2],
    "List 6": [0, 8, 2, 6, 0, 0, 9, 9, 8, 6, 9, 4, 8, 5, 5, 0],
    "List 7": [0, 9, 3, 9, 4, 0, 5, 6, 2, 4, 6, 7, 7, 7, 8, 0],
    "List 8": [4, 9, 1, 4, 8, 5, 1, 4, 4, 7, 4, 9, 3, 9, 6, 7]
}
</Example>

Input: [input_list]
```

Listing 9: GoT Aggregate prompt for Sorting.

```
USER:
<Instruction> Merge the following 2 sorted lists of length [length of
    input1] each, into one sorted list of length [length of input2] using
     a merge sort style approach.
Only output the final merged list without any additional text or thoughts
    !:</Instruction>

<Approach>
To merge the two lists in a merge-sort style approach, follow these steps
    :
1. Compare the first element of both lists.
2. Append the smaller element to the merged list and move to the next
    element in the list from which the smaller element came.
3. Repeat steps 1 and 2 until one of the lists is empty.
4. Append the remaining elements of the non-empty list to the merged list
    .
</Approach>
```

```
Merge the following two lists into one sorted list:
1: [input1]
2: [input2]

Merged list:
```

**GoT on *Document Merging* (DM)**    The prompts for this task were taken from the original implementation of Sorting (Besta et al., 2024a), with minor adjustments. They only contain user messages with the exception of the optimized improve prompt. The prompts used are *Merge* (Listing 10), *Score* (Listing 11), *Aggregate* (Listing 12), *Improve* (original) (Listing 1), and *Improve* (optimized) (Listing 2).

Listing 10: GoT Merge prompt for DM.

```
USER:
Merge the following [N] NDA documents <Doc1> - <Doc[N]> into a single NDA
    , maximizing retained information and minimizing redundancy. Output
    only the created NDA between the tags <Merged> and </Merged>, without
     any additional text.
Here are NDAs <Doc1> - <Doc[N]>:

<Doc1>
[Content of NDA 1]
</Doc1>

<Doc2>
[Content of NDA 2]
</Doc2>

...

<DocN>
[Content of NDA N]
</DocN>
```

Listing 11: GoT Score prompt for DM.

```
USER:
The following NDA  merges NDAs <Doc1> - <Doc[N]>.
Please score the merged NDA  in terms of how much redundant
    information is contained, independent of the original NDAs, as well
    as how much information is retained from the original NDAs.
A score of 10 for redundancy implies that absolutely no information is
    redundant, while a score of 0 implies that at least half of the
    information is redundant (so everything is at least mentioned twice).
A score of 10 for retained information implies that all information from
    the original NDAs is retained, while a score of 0 implies that no
    information is retained.
You may provide reasoning for your scoring, but the final score for
    redundancy should be between the tags <Redundancy> and </Redundancy>,
     and the final score for retained information should be between the
    tags <Retained> and </Retained>, without any additional text within
    any of those tags.

Here are NDAs <Doc1> - <Doc[N]>:

<Doc1>
[Content of NDA 1]
</Doc1>

<Doc2>
[Content of NDA 2]
```

```
</Doc2>

...

<DocN>
[Content of NDA N]
</DocN>

Here is the summary NDA :

[Content of summary]

```

Listing 12: GoT Aggregate prompt for DM.

```
USER:
The following NDAs <S1> - <S[number of summaries]> each merge the initial
    NDAs <Doc1> - <Doc[N]>.
Combine the merged NDAs <S1> - <S[number of summaries]> into a new one,
    maximizing their advantages and overall information retention, while
    minimizing redundancy.
Output only the new NDA between the tags <Merged> and </Merged>, without
    any additional text.

Here are the original NDAs <Doc1> - <Doc[N]>:

<Doc1>
[Content of NDA 1]
</Doc1>

<Doc2>
[Content of NDA 2]
</Doc2>

...

<DocN>
[Content of NDA N]
</DocN>

Here are the summary NDAs <S1> - <S[number of summaries]>:

<S1>
[Content of summary 1]
</S1>

...

<S[number of summaries]>
[Content of summary [number of summaries]]
</S[number of summaries]>
```

**ProbTree on *HotpotQA* and *MuSiQue***   The prompts for this task were taken from the original implementation of Probtree (Cao et al., 2023), with minor adjustments. They only contain user messages. The prompts used are *Understanding* (HotpotQA) (Listing 13), *Understanding* (MuSiQue) (Listing 14), *OpenBook* (Listing 15), *ClosedBook* (HotpotQA) (Listing 16), *ClosedBook* (MuSiQue) (Listing 17), and *ChildAggregate* (Listing 18).

Listing 13: ProbTree Understanding prompt for HotpotQA.

```
USER:
Please generate a hierarchical question decomposition tree (HQDT) with
    json format for a given question. In this tree, the root node is the
```

```
original complex question, and each non-root node is a sub-question
   of its parent. The leaf nodes are atomic questions that cannot be
   further decomposed.
Q: Jeremy Theobald and Christopher Nolan share what profession?
A: {"Jeremy Theobald and Christopher Nolan share what profession?": ["
   What is Jeremy Theobald's profession?", "What is Christopher Nolan's
   profession?"]}.
Q: How many episodes were in the South Korean television series in which
   Ryu Hye-young played Bo-ra?
A: {"How many episodes were in the South Korean television series in
   which Ryu Hye-young played Bo-ra?": ["In which South Korean
   television series Ryu Hye-young played Bo-ra?", "How many episodes
   were <1>?"]}.
Q: Vertical Limit stars which actor who also played astronaut Alan
   Shepard in "The Right Stuff"?
A: {"Vertical Limit stars which actor who also played astronaut Alan
   Shepard in \"The Right Stuff\"?": ["Vertical Limit stars which actor
   ?".

... (in total 16 examples)

Q: [question]
```

Listing 14: ProbTree Understanding prompt for MuSiQue.

```
USER:
Please generate a hierarchical question decomposition tree (HQDT) with
   json format for a given question. In this tree, the root node is the
   original complex question, and each non-root node is a sub-question
   of its parent. The leaf nodes are atomic questions that cannot be
   further decomposed.
Q: When did the first large winter carnival take place in the city where
   CIMI-FM is licensed to broadcast?
A: {"When did the first large winter carnival take place in the city
   where CIMI-FM is licensed to broadcast?": ["Which city is CIMI-FM
   licensed to broadcast?", "When did the first large winter carnival
   take place in <1>?"]}.
Q: What county is Hebron located in, in the same province the Heritage
   Places Protection Act applies to?
A: {"What county is Hebron located in, in the same province the Heritage
   Places Protection Act applies to?": ["Which did Heritage Places
   Protection Act apply to the jurisdiction of?", "which country is
   Hebron, <1> located in?"]}.
Q: What weekly publication in the Connecticut city with the most Zagat
   rated restaurants is issued by university of America-Lite: How
   Imperial Academia Dismantled Our Culture's author?
A: {"What weekly publication in the Connecticut city with the most Zagat
   rated restaurants is issued by university of America-Lite: How
   Imperial Academia Dismantled Our Culture's author?": ["Which
   university was the author of America-Lite: How Imperial Academia
   Dismantled Our Culture educated at?", "What city in Connecticut has
   the highest number of Zagat-rated restaurants?", "What is the weekly
   publication in <2> that is issued by <1>?"], "Which university was
   the author of America-Lite: How Imperial Academia Dismantled Our
   Culture educated at?": ["Who is the author of America-Lite: How
   Imperial Academia Dismantled Our Culture?", "Which university was <1>
   educated at?"]}.

... (in total 15 examples)

Q: [question]
```

Listing 15: ProbTree OpenBook prompt.

```
USER:
Please answer the question and explain why. Output no more than 5 words
    after "So the answer is". End with "So the answer is: <answer>."

#1 Wikipedia Title: First (magazine)
Text: FiRST is a Singaporean movie magazine formerly published monthly,
    now running as a weekly newspaper insert.
#2 Wikipedia Title: Arthur's Magazine
Text: Arthur's Magazine (1844-1846) was an American literary periodical
    published in Philadelphia in the 19th century. Edited by T.S. Arthur,
     it featured work by Edgar A. Poe, J.H. Ingraham, Sarah Josepha Hale,
     Thomas G. Spear, and others. In May 1846 it was merged into "Godey's
     Lady's Book".
#3 Wikipedia Title: First for Women
Text: First for Women is a woman's magazine published by Bauer Media
    Group in the USA. The magazine was started in 1989. It is based in
    Englewood Cliffs, New Jersey. In 2011 the circulation of the magazine
     was 1,310,696 copies.
#4 Wikipedia Title: First Eleven (magazine)
Text: First Eleven is a British specialist magazine for parents of
    children at independent schools.
#5 Wikipedia Title: Earth First! (magazine)
Text: Earth First!, the radical environmental journal, is the official
    publication of the Earth First! movement. First published as a
    newsletter in 1980, it has existed alongside the movement as a way to
     spread commonly held beliefs in "Earth First!" culture, such as
    biocentrism, deep ecology, and direct action. The magazine is also
    commonly known as the "Earth First! Journal".
Q: Which magazine was started first Arthur's Magazine or First for Women?
A: Arthur's Magazine was started in 1844. First for Women was started in
    1989. So Arthur's Magazine was started first. So the answer is:
    Arthur's Magazine.

... (2 more example blocks)

[k retrieved documents]
Q: [question]
A:
```

Listing 16: ProbTree ClosedBook prompt for HotpotQA.

```
USER:
Please answer the question by thinking step-by-step. End with "So the
    answer is: <answer>."
Q: Jeremy Theobald and Christopher Nolan share what profession?
A: Jeremy Theobald is an actor and producer. Christopher Nolan is a
    director, producer, and screenwriter. Therefore, they both share the
    profession of being a producer. So the answer is: producer.
Q: How many episodes were in the South Korean television series in which
    Ryu Hye-young played Bo-ra?
A: The South Korean television series in which Ryu Hye-young played Bo-ra
     is Reply 1988. The number of episodes Reply 1988 has is 20. So the
    answer is: 20.
Q: Vertical Limit stars which actor who also played astronaut Alan
    Shepard in "The Right Stuff"?
A: The movie Vertical Limit starred actors including Chiris O'Donnell,
    Robin Tunney, Scott Glenn, etc. The actor who played astronaut Alan
    Shepard in "The Right Stuff" is Scott Glenn. So the actor who stars
    in Vertical Limit and played astronaut Alan Shepard in "The Right
    Stuff" is Scott Glenn. So the answer is: Scott Glenn.

... (in total 22 examples)

Q:
```

Listing 17: ProbTree ClosedBook prompt for MuSiQue.

```
USER:
Please answer the question by thinking step-by-step. End with "So the
    answer is: <answer>."
Q: When did the first large winter carnival take place in the city where
    CIMI-FM is licensed to broadcast?
A: CIMI-FM is licensed to broadcast in Quebec City. The first large
    winter carnival in Quebec City took place in 1894. So the answer is:
    1894.
Q: When was Neville A. Stanton's employer founded?
A: The employer of Neville A. Stanton is University of Southampton. The
    University of Southampton was founded in 1862. So the answer is:
    1862.
Q: What religion did the black community found?
A: The black community found African Methodist Episcopal Church. So the
    answer is: African Methodist Episcopal Church.

... (in total 23 examples)

Q:
```

Listing 18: ProbTree ChildAggregate prompt for MuSiQue.

```
USER:
Given a qeustion and a context, answer the question and explain why. End
    with "So the answer is: <answer>."

#
Context:
Which famous fashion show Stella Maxwell has been a model for? Victoria's
    Secret.
Since when Victoria's Secret? 1977.

Question:
Stella Maxwell has been a model for a famous fashion shown since when?

Answer:
Stella Maxwell has been a model for a famous fashion shown, Victoria's
    Secret since 2015. So the answer is: since 2015.
#

... (2 more example blocks)

Context:
[subquestions and their answers]

Question:
[question]

Answer:
```

## A.8   USE OF LLMS

For the development of the codebase, Cursor, mainly using OpenAI *GPT-4o*, was used for code completion, documentation, repository structuring, refactoring, and error fixing. No LLMs were used for writing this paper.