# OpenReview forum: "Framework of Thoughts: A Foundation Framework for Dynamic and Optimized Reasoning based on Chains, Trees, and Graphs"
_ICLR.cc/2026/Conference — Submitted to ICLR 2026_

### Official Review · Reviewer_6ww7 · 2025-10-27

**Soundness:** 1
**Presentation:** 1
**Contribution:** 2
**Rating:** 4
**Confidence:** 4

**Summary:**

This paper introduces Framework of Thoughts (FoT), a foundation framework designed to address critical limitations in existing large language model (LLM) reasoning schemes like Chain of Thought (CoT), Tree of Thoughts (ToT), and Graph of Thoughts (GoT). The authors argue that current schemes are often static, requiring manually-defined, problem-specific reasoning structures that lack adaptability. They also suffer from inefficient execution (e.g., sequential, redundant LLM calls) and are often under-optimized in terms of prompts and hyperparameters. FoT is not a new reasoning scheme but rather a framework for building and optimizing other schemes. Its key features include (1) support for dynamic graph structures that can evolve during execution, unlike the static graphs in previous frameworks; (2) faster parallelized execution of operations; (3) intelligent caching (both within a single execution and persistently across samples) to reduce costs and avoid re-execution; and (4) built-in optimization tools for hyperparameters and prompts. To demonstrate its utility, the authors re-implemented ToT, GoT, and ProbTree within FoT. The empirical evaluation shows that FoT's parallelization and caching provided significant runtime accelerations (averaging 10.7x faster) and cost reductions

**Strengths:**

1. The proposed method addresses the problem of static graph structures in prior prompt schemes
2. The proposed method provides efficiency with parallelized execution and caching
3. The proposed method integrates optimization methods.
4. The proposed method shows strong empirical results compared to related works.

**Weaknesses:**

1. The paper presents a limited demonstration of dynamic graphs. The evaluated tasks are all supported by existing methods, indicating that the tasks do not require dynamic graphs. Besides, the dynamic graphs generated by the proposed methods are not provided for qualitative assessments.

2. Although the graphs are claimed to be dynamically built, the type of operations (nodes) seems to be predefined (e.g., split prompt, improve prompt). Thus, the entire framework still requires heavy human engineering.

3. The presentation is poor and does not provide a clear understanding of the proposed method. The method seems to involve dynamically adjusting or growing an execution graph by generating new operation nodes and actually using the graph and compute the operation nodes. However, without a clear explanation or pseudo-code, it is not clear how the entire framework works, what is provided manually, and how the algorithm operates.

**Questions:**

1. Can you provide a clear rundown of how the framework of thought works, including what is manually prepared, and how the algorithm operates after being given all the required input, prompt, and parameter settings?

2. What type of dynamic graphs are created during the experimental tasks, and how do they improve the results compared to fixed graph baseline methods?
3. What additional tasks can the proposed method be applied to? What real-world task would require dynamic graphs and couldn't be solved by prior fix-graph methods?

---

### Official Review · Reviewer_yf4r · 2025-10-30

**Soundness:** 3
**Presentation:** 2
**Contribution:** 2
**Rating:** 2
**Confidence:** 5

**Summary:**

This paper introduces "Framework of Thoughts" (FoT), a foundation framework intended to address systemic limitations in complex LLM reasoning schemes like Chain-of-Thought (CoT), Tree of Thoughts (ToT), and Graph of Thoughts (GoT). The authors identify three primary issues with existing methods: their reliance on static, manually-defined reasoning structures; insufficient optimization of prompts and hyperparameters; and inefficient, sequential execution.

FoT is positioned as a general-purpose system for implementing and running such reasoning algorithms. Its main features include:
1.  **Dynamic Graph Structures**: Modeling reasoning via an "execution graph" that can be modified by operations during runtime.
2.  **Parallel Execution**: A scheduler that attempts to execute independent operations concurrently.
3.  **Caching**: Process-level and persistent caching to reuse results and reduce redundant computations.
4.  **Optimization Hooks**: Integration with external libraries like Optuna and DSPy for automated hyperparameter and prompt optimization.

The authors demonstrate FoT by re-implementing ToT, GoT, and ProbTree. The results show significant improvements in execution speed and cost, and enable performance gains through optimization that would otherwise be computationally expensive.

**Strengths:**

1.  **Clear Problem Identification**: The paper correctly identifies several critical, practical bottlenecks in the implementation of advanced prompting schemes. The focus on systemic issues like static structures, execution inefficiency, and the prohibitive cost of optimization is a relevant and timely contribution to the field.

2.  **Demonstrated Efficiency Gains**: The empirical results effectively showcase the practical utility of the proposed framework. The substantial speed-ups (over 10x in some cases) and cost reductions achieved through parallelization and caching are notable. These efficiency improvements make the process of developing and tuning complex reasoning schemes more accessible and tractable for researchers.

3.  **Useful Open-Source Contribution**: The work culminates in a modular and open framework that could serve as a useful tool for the community. By providing a unified platform for implementing diverse "X-of-Thoughts" schemes, FoT has the potential to facilitate more standardized and reproducible research in this area.

**Weaknesses:**

1.  **Insufficient Substance for a Research Paper; Reads as a Technical Report**: The fundamental weakness of this submission is its lack of research substance. The work is presented as a research paper but its content and contributions are those of a technical report for an early-stage open-source project. It describes the architecture of a software tool that combines a few well-established engineering principles. While potentially useful, it does not propose or validate any new scientific hypotheses, nor does it offer deep insights into the nature of LLM reasoning. The scope and depth of the work are insufficient for a top-tier research conference like ICLR.

2.  **Lack of Novelty in Core Components**: The touted features of FoT are standard, off-the-shelf concepts in software and systems engineering, and their application here is straightforward.
    *   **Optimization**: The paper introduces **no new optimization algorithms**. It merely provides a wrapper around existing, widely-used libraries (Optuna and DSPy). The contribution is one of integration, not innovation. The impact of this "optimization" is therefore limited to the capabilities of these external tools.
    *   **Caching**: Caching is a foundational concept in computer science. The implementation described is a basic key-value store (process-level and persistent), which is the most naive form of caching. There is no research presented on adapting caching for the specific properties of LLM-generated "thoughts" (e.g., semantic caching, approximate matching), which would have constituted a research contribution.
    *   **Parallelism**: The parallelism described is a simple dependency-based task scheduling, a solved problem in distributed computing and workflow management systems.
    *   **Structural Representation**: The claim of enabling "dynamic" graph structures is not sufficiently differentiated from prior art like Adaptive Graph of Thoughts (AGoT)[1]. The paper fails to demonstrate that FoT enables qualitatively new reasoning structures that were previously unattainable.

3.  **Flawed and Unconvincing Experimental Design**: The experimental methodology is not rigorous enough to support strong scientific claims. A major flaw is the use of **different LLMs for different tasks** (GPT-4o, GPT-3.5-Turbo, GPT-4.1-mini). This inconsistent setup makes it impossible to perform a fair comparison of the reasoning schemes (ToT vs. GoT vs. ProbTree), as performance is heavily confounded by the choice of the base LLM. A framework aiming to facilitate fair comparison should, at a minimum, use a controlled experimental setup. This choice undermines the credibility of the evaluation.

4.  **Insufficient Workload and Limited Impact**: The overall workload presented feels preliminary. The paper re-implements only three existing schemes on a handful of tasks. The claimed "dynamic" nature of the framework is not substantially demonstrated, as the implemented schemes (ToT, GoT) are largely static or have limited dynamicism. The paper does not showcase a novel, fully dynamic reasoning agent that would truly stress-test and validate the framework's core design proposition. Consequently, the demonstrated impact of the framework's more advanced features (like dynamic graph modification) is negligible.

[1] Adaptive Graph of Thoughts: Test-Time Adaptive Reasoning Unifying Chain, Tree, and Graph Structures

**Questions:**

1.  **Justification as Research**: Could the authors articulate what they consider to be the primary, non-obvious scientific contribution of this paper, distinct from its engineering contribution of building a software tool? What fundamental hypothesis about AI, LLMs, or reasoning does this work prove or disprove?

2.  **Experimental Control**: Please provide a justification for using different base LLMs across different experimental tasks. How can any valid conclusions be drawn about the relative merits of the reasoning schemes when the most significant variable—the model's capability—is not held constant?

3.  **Scope of Work**: The paper's contribution seems limited to re-implementing a few existing schemes. Was a novel, fully-automatic and dynamic reasoning scheme developed and tested within FoT? If not, why should the community be convinced of the framework's claimed "dynamic" capabilities, which appear to be largely theoretical at this stage?

4.  **Novelty in Optimization**: Beyond acting as a wrapper for existing tools like Optuna and DSPy, does FoT introduce any novel methods or principles for optimizing reasoning graphs? For example, does it leverage the graph structure to perform more efficient prompt or hyperparameter search?

---

### Official Review · Reviewer_G3JY · 2025-10-30

**Soundness:** 2
**Presentation:** 1
**Contribution:** 1
**Rating:** 2
**Confidence:** 5

**Summary:**

This paper introduces Framework of Thoughts (FoT), a foundation framework for implementing and optimizing chain/tree/graph-based prompting schemes (e.g., Tree of Thoughts (ToT), Graph of Thoughts (GoT), ProbTree). FoT aims to address limitations of existing schemes (static structures, suboptimal hyperparameters, inefficient execution) via four core features: dynamic graph structures (evolving during execution), safe parallel execution (with race condition protections), intelligent caching (temporary process cache and persistent cross-sample cache), and built-in optimization (hyperparameters via Optuna, prompts via DSPy). Experiments on 5 tasks (Game of 24 (Go24), Sorting, Document Merging (DM), HotpotQA, MuSiQue) show FoT reduces runtime by up to 35.4× and costs by up to 46% while slightly improving task scores. The authors also release code for reproducibility .

**Strengths:**

* Though not methodologically groundbreaking, FoT integrates fragmented optimizations (dynamic graphs, parallelism, caching, optimization) into a unified, modular framework. It supports diverse operations (LLM calls, tool use, code execution) via Python, lowering the barrier to implementing complex reasoning schemes for developers .

* The work validates FoT on three representative prompting schemes (ToT, GoT, ProbTree) across varied tasks—mathematical reasoning (Go24), structural tasks (Sorting, DM), and multi-hop QA (HotpotQA, MuSiQue)—providing a relatively comprehensive test of its adaptability .

**Weaknesses:**

1. Lack of Methodological Novelty: Core features are either existing capabilities or off-the-shelf integrations.
  * Dynamic graphs: LangGraph (2024) already enables adding/removing nodes/edges during execution; FoT’s separation of “execution graph” and “reasoning graph” is a cosmetic distinction, not a functional innovation .
  * Optimization: FoT directly uses Optuna (Akiba et al., 2019) for hyperparameters and DSPy’s COPRO (Khattab et al., 2023) for prompts—no custom algorithms or adaptive strategies are proposed .
  * Parallelism/caching: GoT (Besta et al., 2024a) already supports parallel sublist merging, and LangChain’s async modules enable concurrent calls; FoT’s “safe parallel constraints” (e.g., limiting modifications to exclusive descendants) are trivial adjustments.

2. Limited Benchmarks: All experiments focus on structured/semi-structured tasks (math, sorting, merging, multi-hop QA) with no unstructured tasks (e.g., code debugging, logical puzzles, creative writing)—tasks where dynamic graphs might add value, leaving FoT’s “general-purpose” claim unproven. Datasets are undersized: Sorting/DM only have 50 test instances each; HotpotQA/MuSiQue use 1,000 test instances (far fewer than standard benchmarks like HotpotQA’s official 10k+ test set)—results may lack statistical significance .

3. Inadequate Baseline Comparisons: FoT only compares to naive baselines (sequential execution, no cache) and the original schemes (ToT, GoT, ProbTree)—no comparisons to state-of-the-art dynamic frameworks.

**Questions:**

See the weakness part.

---

### Official Review · Reviewer_5shY · 2025-11-01

**Soundness:** 2
**Presentation:** 3
**Contribution:** 2
**Rating:** 4
**Confidence:** 4

**Summary:**

This paper proposes Framework of Thoughts (FoT), a general-purpose framework to implement and optimize multi-prompt reasoning schemes (e.g. CoT/ToT/GoT/ProbTree). The key features of FoT are: 1) dynamic execution graphs that may evolve during inference, together with a reasoning graph that generate thoughts for reasoning; 2) safe parallel scheduling under graph-modification constraints; 3) caching to reduce repeated LLM calls. The paper re-implemented ToT, GoT, and ProbTree in FoT and evaluate across Game of 24, Sorting, Document Merging, HotpotQA etc. They report large speedups from parallelization and cost reductions from caching and modest task-score gains.

**Strengths:**

- The proposal and distinction between execution and reasoning graphs, and the definition of allowable graph mutations (ancestors/descendants/exclusive descendants) during parallel execution, is well-motivated and easy to reason about.
- Parallel scheduling plus two-tier caching is practical and helpful, especially for optimization loops where repeated calls dominate runtime.

**Weaknesses:**

Novelty is more infrastructural than algorithmic. Much of FoT bundles known ideas (parallel execution, caching, hyperparameter, prompt tuning) into a framework; the new scientific insight beyond system engineering is limited. The paper's dynamic graph proposal is interesting, but the story overlaps with existing orchestration frameworks (e.g., LangGraph execution DAGs, parallel branches, caching). The delta feels more engineering consolidation than a conceptual leap.

**Questions:**

- Do the dynamic graph generation procedure ensure any formal properties (e.g., determinism of final reasoning graph given fixed seeds)? If not then the graph topology would be uncontrollable, not sure what failure modes did you observe and how are they mitigated?

- I'm curious to know how the dynamically generated graphs differ from predefined graphs (trees, graphs etc.). Are the dynamic graphs usually smaller or shallower than the predefined graphs? How would you justify the dynamically constructed graph topolgies are more suitable for the problems to be solved?

- How are the cache keys defined? Is it something like prompts + inputs + model generations + tool specifications?

---

### Author Response · Authors · 2025-12-04
**Response to Reviewers and Area Chair (Part 1)**

We thank all reviewers for the time and thought they invested in evaluating our submission. While the reviews raise fair questions, we believe they do not fully reflect the central contribution of our work and the value it offers to the community—especially in the context of the "infrastructure, software libraries, hardware, systems" primary area under which we submitted.

## FoT as a Research Contribution in the Infrastructure Area

A recurring theme across the reviews is that while the empirical gains of FoT are clear, its individual components may not appear novel in isolation (e.g., “Novelty is more infrastructural than algorithmic” by Reviewer 5shY). We agree that many of the ideas underlying FoT—parallel execution, caching, optimization—exist independently. The contribution of FoT is **not** in reinventing any one of these mechanisms; it is in **integrating them into a single, coherent, efficient-by-default execution model and framework** for LLM-based reasoning schemes.

Schemes such as ToT, GoT, and ProbTree are conceptually compelling but are often implemented in ways that repeat identical computation, serialize operations unnecessarily, and make systematic optimization nearly impossible. FoT demonstrates that **a unified abstraction paired with strong systems design can dramatically improve runtime, cost, and even task scores** without modifying the underlying reasoning logic. In other words, FoT is intended to be “more infrastructural than algorithmic”.

Importantly, reviewers themselves acknowledge the magnitude of these improvements (e.g., “FoT reduces runtime by up to 35.4× and costs by up to 46% while slightly improving task scores” by Reviewer G3JY). These gains—order-of-magnitude speedups, substantial cost reductions, and non-trivial accuracy boosts across diverse tasks—are not the result of new algorithms, but the result of a more principled execution model. In this sense, FoT is a **foundational enabler** and our paper exposes the latent potential already present in widely-used reasoning schemes and provides the community with a vehicle for scalable experimentation, iteration, and future research.

We believe that this is precisely the type of contribution that fits the **infrastructure track**: not a new reasoning algorithm, but a platform that makes existing and future reasoning algorithms far more powerful, tractable, comparable, and reusable.

## On Dynamic Graphs and the Value of FoT’s Abstraction

Reviewers questioned how strongly the experiments demonstrate the benefits of FoT’s support for dynamic reasoning graphs. We would like to clarify that:

1. **The evaluated schemes already include substantial dynamism.** ToT dynamically expands, filters, and prunes search trees based on learned assessments. ProbTree contains significant dynamic structure, including tree growth through dynamic question decomposition, selection of answer paths, and the integration of retrieval steps. These are not static pipelines; ToT, and even more ProbTree, are already representatives of the type of dynamic, branching, data-dependent processes a framework like FoT is designed to support.

2. **We intentionally reimplemented well-known, well-cited, and peer-reviewed schemes** rather than newer, not yet validated ones (e.g., AGoT), to ensure scientific credibility. We believe that demonstrating large efficiency and performance gains on established benchmarks is a stronger and more reliable indicator of FoT’s value.

3. **Theoretical value exceeds what any single experiment could show.** For future reasoning schemes to exhibit agency, graphs must be dynamic, as the choice of next operations reflects this agency. FoT is the first framework to make fully dynamic reasoning execution a first-class concept, with clear semantics for how operations can modify the graph during execution while preserving correctness and enabling safe parallelism. This is a necessary precondition for a future in which reasoning schemes exhibit greater agency—where prompts and operations can themselves be adaptively generated by the LLM at runtime.

4. **FoT does not require heavy human engineering.** While our reimplementations use existing human-designed operations and prompts (to maintain comparability with original papers), nothing in FoT’s design prevents future schemes from using simpler, more general-purpose operations combined with dynamic LLM-driven decision-making. In fact, users can implement an operation in FoT that uses an LLM to define further task-specific operations and their prompts, thereby creating a general-purpose reasoning scheme. Enabling such next-generation, self-directed reasoning schemes is part of the motivation behind developing FoT’s abstractions.

Thus, although our experiments focus on existing schemes, the framework is intentionally designed for a broader scope: dynamic, adaptive, general-purpose reasoning systems—something for which FoT serves as a critical infrastructural foundation.

---

### Author Response · Authors · 2025-12-04
**Response to Reviewers and Area Chair (Part 2)**

## On Experimental Design and Choice of Tasks, Baselines, and LLMs

Some reviewers raised questions regarding our evaluation design. We would like to clarify that our evaluation follows clear scientific reasoning:

- We reimplemented **three structurally diverse, widely-cited reasoning schemes**—ToT, GoT, and ProbTree.

- We evaluated them on the **tasks used in their original papers**: Go24 (symbolic search), Sorting (algorithmic), Document Merging (document-level synthesis), HotpotQA and MuSiQue (multi-hop QA).
These collectively cover tree and graph structures, both static and dynamic, and both synthetic and real-world-like tasks.

- The purpose was **not** to compare the schemes against each other, nor to introduce new reasoning algorithms, but to show what FoT contributes **relative to the original implementations** of these schemes.

- For the same reason, we used the **LLMs most faithful to each original paper**, to ensure that any differences in performance were due to FoT—not to a stronger or weaker underlying LLM. Our baselines were faithful reimplementations, not strawmen. We had to make some deviations, where LLMs were no longer available or too costly for our budget. We transparently documented these deviations in section 4 of our paper.

This design allowed us to isolate and measure FoT’s improvements fairly and directly:
the same reasoning schemes, same logic, same models—just executed under a more principled, efficient framework. The fact that FoT consistently yields large runtime and cost reductions and often improves task scores under these controlled conditions speaks strongly to the framework’s practical value.

## Conclusion

Despite the reviews’ concerns about novelty, we respectfully submit that FoT represents a meaningful and timely contribution to the ICLR community—particularly within the infrastructure area. It offers:

- a **unified abstraction** for dynamic LLM reasoning,

- a **sound execution model** that supports graph evolution, safe parallelism, and caching,

- a **single framework** that operationalizes these ideas “efficient-by-default,”

- **significant empirical gains** across multiple well-known reasoning schemes,

- and the **infrastructural foundation** enabling the next generation of adaptive reasoning systems.

The reviews acknowledge the practical improvements FoT delivers. We believe it would be a loss to overlook a contribution that not only exposes the untapped efficiency and performance potential of existing reasoning schemes but also lays the foundation for more general and more powerful reasoning systems to come.

We hope that FoT will be viewed in this light—as a valuable infrastructural and conceptual contribution that meaningfully advances the practice of LLM-based reasoning research and a call to the community not to ignore cost and runtime of LLM-based reasoning schemes.

---

### Meta-Review · Area_Chair_HRgb · 2026-01-07

**Summary:**

This paper introduces the Framework of Thoughts, a software framework aimed at implementing and optimizing structured reasoning schemes for large language models such as Tree of Thoughts. While reviewers acknowledged the practical utility and efficiency gains demonstrated through speedups and cost reductions, significant concerns were raised regarding the core novelty of the work, with multiple reviewers noting its contributions are primarily infrastructural, integrating existing concepts like parallel execution and caching rather than offering new algorithmic insights. Further criticisms highlighted a lack of rigorous experimental validation, insufficient demonstration of the promised dynamic capabilities, and a presentation that obscures the framework's operational mechanics. Given these fundamental issues pertaining to scientific contribution and methodological rigor, the paper in its current form is not suitable for acceptance.

**Reviewer Concerns:**

The authors' rebuttal did address some specific methodological questions. They clarified that their experimental design used different LLMs to maintain faithfulness to the original baselines, which responds to concerns about unfair comparisons. They also provided a more detailed explanation of the dynamic elements present in the implemented schemes like Tree of Thoughts, offering a justification for the framework's support of evolving graphs.

However, the most significant concerns regarding the paper's contribution remain outstanding. The central criticism that the work primarily integrates established engineering concepts rather than presenting a novel research insight persists. The rebuttal did not sufficiently demonstrate how the framework's dynamic capabilities enable reasoning structures meaningfully beyond prior art, nor did it provide the clearer algorithmic exposition or formal guarantees requested. The overarching assessment that the contribution is more infrastructural than scientific, and that the experimental validation does not fully substantiate the claimed generality, is not effectively countered.

**Reviewer Scores:**

| Reviewer | Initial Score | Predicted New Score | Reasoning |
| - | -| - | -|
| yf4r | 2 | 2 | The rebuttal clarifies the experimental design but does not address the core criticism that the work lacks scientific novelty and reads as a technical report. The reviewer's fundamental concern about contribution remains unchanged. |
| G3JY | 2  | 2  | The authors' response does not counter the central weaknesses regarding methodological novelty or limited benchmark scope. The reviewer's initial assessment of insufficient contribution is likely to stand. |
| 5shY | 4  | 4  | The reviewer already noted the infrastructural novelty. The rebuttal reinforces the practical utility, which might align with the reviewer's view, but is unlikely to elevate the score given the acknowledged lack of algorithmic novelty. |
| 6ww7 | 4  | 4  | While the rebuttal offers some explanation on dynamism, it does not fully satisfy the requests for clearer algorithmic exposition or a qualitative demonstration of dynamic graphs. The original concerns about presentation and demonstration persist. |

---

### Decision · Program_Chairs · 2026-01-26

Reject